# Accurately Identifying Cerebroarterial Stenosis from Angiography Reports Using Natural Language Processing Approaches

**DOI:** 10.3390/diagnostics12081882

**Published:** 2022-08-03

**Authors:** Ching-Heng Lin, Kai-Cheng Hsu, Chih-Kuang Liang, Tsong-Hai Lee, Ching-Sen Shih, Yang C. Fann

**Affiliations:** 1Center for Artificial Intelligence in Medicine, Chang Gung Memorial Hospital, Taoyuan 33305, Taiwan; chingheng113@gmail.com; 2Bachelor Program in Artificial Intelligence, Chang Gung University, Taoyuan 33305, Taiwan; 3Bioinformatics Section, National Institute of Neurological Disorders and Stroke, National Institutes of Health, Bethesda, MD 20892, USA; edwardfirst@gmail.com (K.-C.H.); ck.vghks@gmail.com (C.-K.L.); 4Department of Medicine, China Medical University, Taichung 40447, Taiwan; 5Artificial Intelligence Center for Medical Diagnosis, China Medical University Hospital, Taichung 40402, Taiwan; 6Department of Neurology, China Medical University Hospital, Taichung 40402, Taiwan; 7Center for Geriatrics and Gerontology, Kaohsiung Veterans General Hospital, Kaohsiung 81362, Taiwan; 8Division of Neurology, Department of Medicine, Kaohsiung Veterans General Hospital, Kaohsiung 81362, Taiwan; csshih@vghks.gov.tw; 9Aging and Health Research Center, National Yang Ming University, Taipei 11221, Taiwan; 10Stroke Center and Department of Neurology, Chang Gung Memorial Hospital, Linkou Medical Center, Taoyuan 33333, Taiwan; thlee@cgmh.org.tw; 11College of Medicine, Chang Gung University, Taoyuan 33302, Taiwan

**Keywords:** intracranial artery stenosis, cerebrovascular diseases, natural language processing, ruled-based model, deep learning

## Abstract

Patients with intracranial artery stenosis show high incidence of stroke. Angiography reports contain rich but underutilized information that can enable the detection of cerebrovascular diseases. This study evaluated various natural language processing (NLP) techniques to accurately identify eleven intracranial artery stenosis from angiography reports. Three NLP models, including a rule-based model, a recurrent neural network (RNN), and a contextualized language model, XLNet, were developed and evaluated by internal–external cross-validation. In this study, angiography reports from two independent medical centers (9614 for training and internal validation testing and 315 as external validation) were assessed. The internal testing results showed that XLNet had the best performance, with a receiver operating characteristic curve (AUROC) ranging from 0.97 to 0.99 using eleven targeted arteries. The rule-based model attained an AUROC from 0.92 to 0.96, and the RNN long short-term memory model attained an AUROC from 0.95 to 0.97. The study showed the potential application of NLP techniques such as the XLNet model for the routine and automatic screening of patients with high risk of intracranial artery stenosis using angiography reports. However, the NLP models were investigated based on relatively small sample sizes with very different report writing styles and a prevalence of stenosis case distributions, revealing challenges for model generalization.

## 1. Introduction

Intracranial arterial (cerebroarterial) stenosis (IAS), which affects the middle cerebral artery, the intracranial portion of the internal carotid artery, the vertebrobasilar artery, and the posterior and anterior cerebral arteries, is a common risk factor for ischemic stroke, especially in the Asian population [1]. The prevalence of IAS for stroke patients is around 33%–50% in China, Thailand, Singapore, South Korea, and Japan, which is higher than that among Caucasian populations (20%) [2,3]; the prevalence of asymptomatic intracranial arterial stenosis has been estimated to be around 4.0%–13.7% [4,5]. Optimal treatments for intracranial arterial stenosis remains challenging due to racial differences, distinctive stroke etiologies, and diverse locations of IAS; however, high-risk patients might benefit from preventive medical therapy if the risk is detected early [6]. Nowadays, various imaging modalities, including transcranial Doppler, magnetic resonance angiography (MRA), and computed tomography angiography, had been routinely used to screen intracranial arterial stenosis. But in real-world clinical practices, it is time consuming and resource intensive when unstructured reports were used to extract various health conditions or detect diseases thus remained largely unutilized. It has been estimated that about 80% of medical data, including reports, remain unstructured which used in real-world clinical applications (e.g., information extraction for disease risks identification) [7].

Recent studies utilizing machine learning (ML) and natural language processing (NLP) techniques to improve the performance of clinical report information extraction have been reported with varied results [8,9,10,11]. With high prevalence of arterial stenosis in the Asian population, there is an urgent need for effective approaches for detecting and preventing potential risks of stroke. ML/NLP techniques have shown to be promising in their conversion of free text input into structured data to enable the automatic identification and extraction of information and knowledge from unstructured medical reports. We aimed to investigate different NLP approaches to ascertain which model is the most effective in allowing healthcare physicians to effectively and accurately identify complex, multilabel artery stenosis from commonly available angiography reports. In addition, to properly evaluate and validate effective NLP techniques that can potentially be used to develop clinical applications in real-world hospital settings, we employed a large cohort from a national medical center and a selected external dataset from a regional medical center to better assess our model’s performance and its potential generalizability. In this study, we surveyed the most representative classic rule-based methods and state-of-the-art deep learning NLP methods for evaluation and comparison. These included a rule-based model with a handcrafted feature-based approach, a long short-term memory (LSTM) machine learning model with a recurrent neural network approach, and an XLNet deep learning method with a pretrained language model approach.

## 2. Materials and Methods

### 2.1. Data Collection and Preprocessing

This study employed two different data sources for model training, testing, and validation. The primary source was the Linkou Chang Gung Memorial Hospital (LCGMH), one of the largest medical centers in Taiwan, which we utilized for internal training and testing. The other external dataset, used for testing, was drawn from our collaboration site of Kaohsiung Veterans General Hospital (KSVGH), an independent local medical center. In the LCGMH dataset, patients who underwent cerebral angiography and color duplex ultrasound at LCGMH from January 2007 to December 2016 were enrolled and included. From the KSVGH dataset, those participants who were admitted due to acute ischemic stroke between July 2018 and June 2019 and had reports of MRA were recruited in the external testing dataset to verify the performance of the stenosis identification models.

This study collected a total of 9614 reports of magnetic resonance angiography (MRA) from the LCGMH as the training and internal testing dataset, with a limited number of 315 MRA reports which were approved by KSVGH IRB and were used as the external validation dataset. Examples of angiography reports from two hospitals with labels can be found in the Appendix A. All the angiography reports from two hospitals were preprocessed by removing Chinese characters/sentences, special characters, and extra spaces prior to model training. To properly validate the stenosis, two chief neurologists from both hospitals independently labeled and validated the stenosis degree (<50% or ≥50% diameter stenosis) of 11 target arteries, which are routinely used for the diagnosis of intracranial artery stenosis in angiography reports (both LCGMH and KSVGH datasets). The 11 target arteries used were the following: left/right intracranial internal artery (LIICA, RIICA, respectively); left/right anterior cerebral artery (LACA, RACA, respectively); left/right middle cerebral artery (LMCA, RMCA, respectively); left/right posterior cerebral artery (LPCA, RPCA, respectively); left/right intracranial vertebral artery (LIVA, RIVA, respectively); and the basilar artery (BA). The degree of arterial stenosis was determined according to the North American Symptomatic Carotid Endarterectomy Trial (NASCET) criteria [12]. For easy comparison, the percentages of cases with confirmed stenosis for each artery, used in both internal and external datasets, are provided in Table 1. The ground truth for the evaluation was established by a consensus between the two chief neurologists, who shared similarities in medical training and years of practice. They evaluated each angiography report and reached a consensus, as with the process that is performed in real-world hospital settings. In addition, with selected keywords in the angiography reports, such as significant stenosis, tight stenosis, severe stenosis, and high-grade stenosis, the consensuses can be readily reached and recorded within rule-based guidelines (Appendix A Appendix A).

### 2.2. Stenosis Identification Models

To develop and assess the best NLP model for the task of intracranial artery stenosis identification, with the aim of improving patient care, three NLP algorithms (a classic rule-based NLP approach, a LSTM recurrent neural network, and a contextualized language XLNet model) were evaluated for their ability to accurately identify stenosis risks from angiography reports, as shown in Figure 1.

#### 2.2.1. Rule-Based NLP Model

The rule-based NLP algorithm was developed based on expert-derived keywords and rules that were handcrafted by medical experts [13]. It has been widely adopted and reported for a variety of radiology NLP tasks [14,15,16]. As shown in Figure 1 panel A, a set of identification rules that was created by neurologists, containing reference keywords, matching rules, and exclusion criteria for negated findings (see Appendix A Appendix A), was developed. This set of rules was then transformed into a series of regular expression rules by R package stringr, available at https://cloud.r-project.org/package=stringr (accessed on 23 January 2022). The rule-based model also incorporated a standard NLP pipeline, including section and sentence segmentations to increase its performance. The reports obtained from LCGMH mainly contain four sections, as follows: basic patient information, clinical information, image findings, and impression sections. Those from KVGH contained similarly titled sections, as follows: basic patient information, methods, findings, and impressions. We chose two relevant sections validated by physicians to auto-extract the contents of the image findings and impression sections from two hospitals’ reports for the NLP identification tasks. The extracted paragraphs from the target sections were further processed and evaluated for sentence segmentation by the R package spacyr, available at https://spacyr.quanteda.io (accessed on 23 January 2022). 

#### 2.2.2. Long Short-Term Memory Model

The LSTM model is a recurrent neural network architecture that has been designed to address data sequences of varying length and to capture long-term dependencies [17]. This study built a deep LSTM model for stenosis identification using Keras v2.3.1 [18] with the TensorFlow v2.1.0 [19] backend. The model structure is shown in Figure 1, panel B, and consists of one embedding layer with an output dimension of 128, two LSTM layers of 64, 32 cells, and the fully connected output layer corresponding to the number of target arteries. The total number of parameters used was 1,275,697; to reduce overfitting we applied dropout with a probability of 0.2 to each layer. Since a patient may have several artery sections mentioned in an angiography report, the sigmoid activation function was adopted in the output layer to help solve the multilabel text classification problem. The binary cross-entropy loss function and Adam optimizer [20] of 1 × 10^−3^ learning rate were selected for an optimal model training.

#### 2.2.3. XLNet Model

The XLNet model was a contextualized self-attention-based language model which combined the advantages of autoregressive and autoencoder methods by using permutation language modeling techniques. It used the Transformer-XL model [21] to make long-text encoding more effective. In a recent report, XLNet was shown to overcome the limitations of BERT [22] and achieved state-of-the-art results in several NLP tasks [23]. The XLNet NLP model was pretrained on various text corpora including BooksCorpus [24], English Wikipedia, Giga5 [25], ClueWeb 2012-B [26], and Common Crawl [27]. In this study, an XLNet model was developed from pretrained weights provided by the HuggingFace’s Transformers library [28], with default parameters of 24 hidden layers in the Transformer encoder, 16 attention heads for each attention layer, and 340 M parameters. As shown in Figure 1 panel C, the token embedding layer transformed the texts to token identifiers and fed them into an XLNet pretrained model. This contained a linear layer corresponding to the number of target arteries employed in this study, with a sigmoid activation function added on top of the contexture embedding layer. For fine-tuning the model, we selected AdamW optimizer [29] with 1 × 10^−5^ learning rate and the sigmoid cross entropy with logits as the loss function. The final model was then trained for 20 epochs with 16 batch sizes.

### 2.3. NLP Model Assessments

To assess the identification capabilities and performance of three NLP models, this study employed a 10-time hold-out cross-validation. As shown in Figure 2, for each validation round, we used 80% of the data (7692 cases) from the internal LCGMH dataset for the training of three models, and the remaining 20% of the LCGMH dataset (1922 cases) was used for internal testing and validation. To further investigate the generalizability of our models to real-world clinical applications, we obtained a limited IRB-approved KSVGH dataset (315 cases) as an independent external testing dataset that was included in neither the model training nor the rule-based model development. Due to the smaller sample size, with an imbalanced distribution of positive cases (i.e., stenosis) of each artery (as shown in Table 1), to best evaluate the models’ performance we used the well-accepted measure of area under the receiver operating characteristic curve (AUROC). This approach was selected for its specificity and sensitivity. If the predictions were not derived from probabilities, then the AUROC was acquired from a confusion matrix, as follows: 1/2 [(tp/(tp + fn)) + (tn/(tn + fp))]; Specificity is tn/(tn + fp), and Sensitivity is tp/(tp + fn), where TP, TN, FP, and FN denoted true positives (i.e., the model correctly identified the stenosis), true negatives (i.e., the model correctly identified no stenosis), false positives (i.e., the model incorrectly identified the stenosis), and false negatives (i.e., the model incorrectly identified no stenosis), respectively.

## 3. Results

The comparison of AUROC results between internal and external testing datasets for the intracranial artery stenosis identification task from three different NLP models is summarized in Table 2 and the receiver operating characteristic curves shown are in Figure 3. For the internal testing dataset from LCGMH, the best performance was obtained by the XLNet model with an AUROC ranging from 0.97 ± 0.01 RPCA to 0.99 ± 0.00 RIVA among all 11 target arteries evaluated. The rule-based model which we considered as a performance benchmark achieved its best AUROC in RIVA (0.96 ± 0.01) and its worst AUROC in BA (0.92 ± 0.02). XLNet, built with a pretrained model developed in this study, clearly demonstrated its superior performance in detecting stenosis with consistently higher performance across all target arteries compared with those of LSTM model. 

The performance of the three stenosis identification models dropped significantly with the smaller external testing dataset obtained from KSVGH. Although the XLNet model still outperformed both the rule-based model and the LSTM model in all target arteries in this external testing dataset, the range of AUROC declined and varied greatly from 0.79 ± 0.12 LPCA to 0.99 ± 0.01 LACA, as shown in Table 2. In addition, the performance of the XLNet model was found to increase on those arteries with higher prevalence of stenosis. The LSTM model resulted in better detection of stenosis overall compared with the rule-based model, except in few arteries (e.g., RPCS, RIVA, and BA) in the external dataset; however, both were found to be less than optimal compared with the XLNet model. 

Compared with previously reported studies, the stenosis-identification task employed in our study was a complicated multilabel challenge of identifying and reporting up to 11 target arteries labeled with their individual stenosis statuses and lesion locations. To further compare the performance of our XLNet model with other previous studies, we binarized the angiography reports as being with/without stenosis and tested the identification ability. The results showed that XLNet presented an improved performance when the identification task was simplified, with AUROC = 0.98 ± 0.0 and AUROC = 0.97 ± 0.01 from the internal and external testing datasets, respectively. For more evaluation metrics such as sensitivity and specificity, as well as a detailed AUROC of the XLNet model with different training epochs and learning rates in each stenosis detection task, see Appendix A Appendix A, as described in Materials and Methods.

## 4. Discussion

In this study, we evaluated three different approaches of NLP models and demonstrated that machine-learning-based NLP techniques are more efficient in identifying multilabel intracranial arterial stenosis and its location from angiography reports. As summarized in Table 2, the XLNet model outperformed the rule-based and LSTM models in the internal training and testing assessments of the stenosis-identification task. It should be noted that, although machine learning approaches dominated the current NLP research, the rule-based approach still has several advantages, such as its minimal requirement for labeled data. This was demonstrated by its performance with the large dataset employed in our study. In this case, only a set of handcrafted rules with a list of expert-derived keywords were needed to develop the detection and prediction algorithms, in contrast to the machine learning methods (e.g., LSTM and XLNet) that required a large labeling dataset, not only for model training but also for the evaluation and validation tasks. In addition, in the rule-based model, it is easy to incorporate domain knowledge for general improvement and interpretability through its declarative expressions. Unlike machine learning approaches that demand relatively high computing resources to train a model, a rule-based model does not require a training phase. Our internal testing results showed that a well-crafted rule-based model has a reasonable ability to identify stenosis from the reports. Therefore, with insufficient annotated datasets or a lack of computing resources, rule-based models can serve as an alternative solution. However, it should be noted that factors such as ambiguity in terminology or findings that may be too complex for clinicians to interpret and craft meaningful keywords, may limit and impact rule-based NLPs’ performance. In addition, to further improve a rule-based model, appropriate preprocessing steps including spell checking, word stemming, and removal of meaningless words could be helpful for improving the performance [30]. 

The present study also discovered that dataset inconsistency and different writing styles most significantly impacted the rule-based model, since it strongly depended on the keywords, the structure of sentences, abbreviations, and term variations in the reports. The performance results of the three developed stenosis identification models showed a decline in performance when using the external KSVGH testing dataset. We discovered that the main reasons for the poor performance were the different writing styles and the medical training in report writing practices of physicians in different hospitals. For example, one angiography report from the internal testing/validation dataset of LCGMH mentioned “Total occlusion of right distal VA”, but did not point out or describe any normal artery. Conversely, those from the external testing dataset (i.e., KSVGH) had more detailed descriptions, as follows: “MRA shows no evidence of occlusion or high-grade stenosis over intracranial portion of the internal carotid artery and vertebral basilar artery, and main trunk of the bilateral anterior cerebral arteries and middle cerebral arteries and posterior cerebral arteries”. In addition, similar but different terms used, such as “stenosis” and “occlusion” in the internal testing dataset versus “paucity” in the external testing dataset, also contributed to the performance variations in these models. Furthermore, same diagnoses with different descriptions, such as “Total occlusion of right ICA” versus “Distal internal carotid artery (ICA): severe stenosis, right side”, further complicated the validation tasks between the two hospitals’ datasets. Descriptions of the same lesion location were also found to be quite different. It was also noted that the reports from LCGMH tended to describe stenosis conditions separately, and placed location descriptions before each artery, e.g., “Stenosis of right MCA M1-2, left PCA P1 and right PCA P2-3 junction (all > 50%)”, but those from KSVGH would write “High grade stenosis over bilateral posterior cerebral arteries” or “Posterior cerebral artery (PCA): moderate to severe stenosis, left”, for example. The LSTM model, which does not have pretrained language representation, was also greatly affected by the complicated context forms described above. Challenges in most ML modeling research, such as dataset shift (e.g., an imbalanced dataset), were also found in this study, with changes in data distribution or prevalence [31]. As shown in Table 1, the prevalence of stenosis from each artery was found to be very different between internal and external datasets, a disparity that is often observed in many real-world cases with limited datasets obtained from IRB-approved data for ML-based studies and with population bias represented in different regional hospitals. In addition, the number of positive (i.e., has stenosis) cases versus negative (i.e., no stenosis) cases was also found to be imbalanced; therefore, there were further challenges in developing a generalized clinical ML-based NLP tool. Tackling this dataset shift problem to generate better ML-based NLP models could involve the application of continual learning or federated learning, which enables a model to learn continually from a stream of data from participating hospitals. This was shown to make models more robust and less susceptible to changes in data distributions [32]. Despite these data challenges in the external testing dataset, this study clearly demonstrated that the XLNet-based NLP model can produce superior performance in detecting stenosis from clinical reports with large training and testing dataset from within same hospital; therefore, this model can help streamline and improve physicians’ clinical routines. 

Among the three different approaches implemented and evaluated in this study, XLNet clearly surpassed other two NLP models. Recent studies have demonstrated that pretrained language models are highly successful in many NLP tasks [33], with XLNet being one of the architectures reported. Pretraining in an unsupervised manner on a large medical report text corpus can enable models to learn universal language representations, aiding downstream tasks, as clearly reflected in the model’s performance with our internal testing dataset. Our results when the XLNet model performed the simple downstream task indicated that the pretrained language model approach provided a better model initialization step, and reduced the problem of model generalizability compared with the other NLP models [34]**.** In general, the size of model parameters could contribute to better performance [35], as demonstrated in our study; the number of parameters used for XLNet was 340 times larger than that for LSTM (340 M vs. 1 M). Several recent studies explored different computational approaches for simple disease-identification tasks from a radiology report. For example, Wu et al. proposed three machine learning models to identify patients with the conditions of intracranial artery stenosis based on their ultrasound report, including a logistic regression model, a field-aware convolution neural network, and a recurrent neural network (RNN) with an attention mechanism. Their results showed that the RNN–attention model achieved only approx. 95.4% accuracy [36]. Drozdov et al. also evaluated thirteen supervised classifiers in identifying normal, abnormal and unclear chest radiography reports. Their bidirectional long short-term memory networks with an attention mechanism effectively identified only three different types of chest radiography reports in the internal testing dataset, with an f1 score of 0.94, and in the external testing dataset, with an f1 score of 0.90 [37]. Those studies were performed on much simpler identification tasks compared with the multilabel challenges tackled in our study, with less favorable performances. 

Identifying not only the stenosis status but also its accurate lesion location from large angiography datasets remains challenging, but it is critical for real-world clinical applications of identifying the risks of potential stroke. Depending on each hospital setting, this study provided a clear pathway with three different NLP approaches to best ascertain and tackle this challenge. Our future work will focus on obtaining more datasets from different hospitals and employing different or hybrid methodologies such as federated learning to further assess and improve stenosis identification models. In general, NLP models pretrained for general-purpose language understanding have shown poor performance in specific-domain NLP tasks [38,39,40]. Huang et al. proposed a domain-specific pretrained model called Clinical XLNet [41]. These proposed approaches and models will be explored in the future to further improve stenosis-identification tasks.

## 5. Conclusions

In this work, we evaluated three very different NLP approaches in the aim of finding the best model for automating and accurately identifying potential stenosis risks on 11 targeted arteries from angiography reports. The results clearly demonstrated that a contextualized language XLNet model has superior performance compared with LSTM and rule-based models. This ML approach can greatly assist physicians, by alleviating the requirement for the manual human interventions which are currently employed in hospital settings. In addition, the information extracted from the XLNet model in this study can be combined with other clinical risk factors to assist physicians in providing better preventive stroke care, which will improve patient health.

## Figures and Tables

**Figure 1 diagnostics-12-01882-f001:**
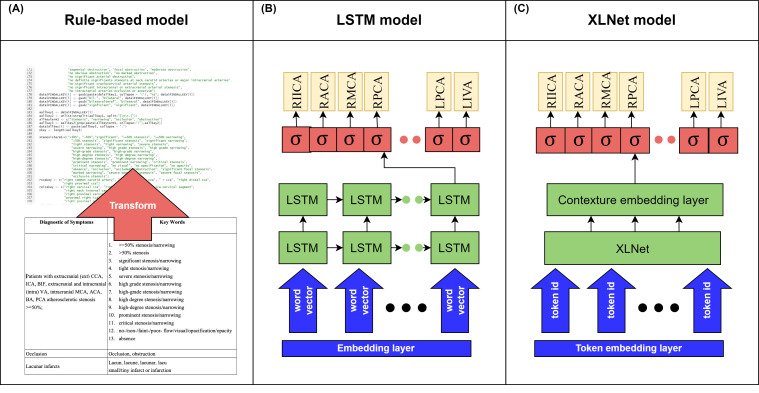
Overview of stenosis identification models. Panel A presents a rule-based model, which is a handcrafted feature-based approach. Panel B presents a long short-term memory model, which is a recurrent neural network approach. Panel C presents XLNet, which is a pretrained language model approach.

**Figure 2 diagnostics-12-01882-f002:**
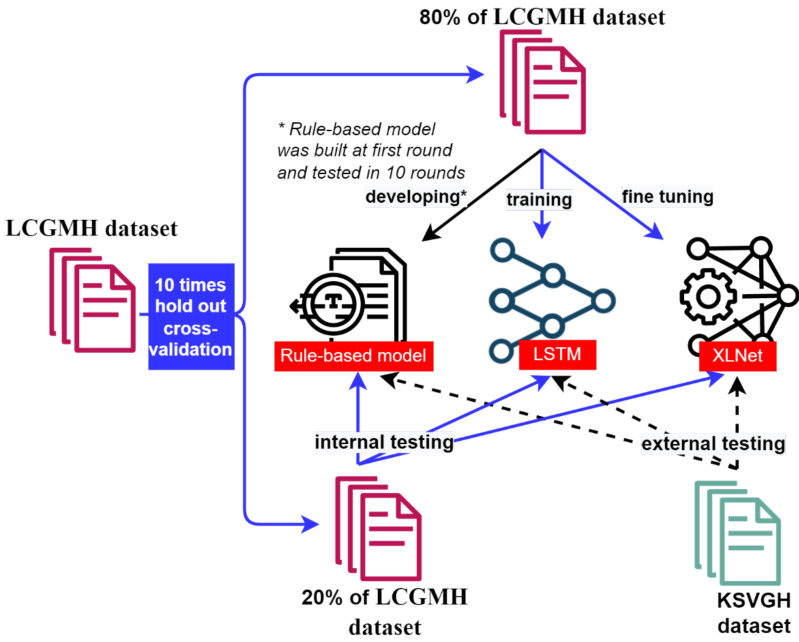
The process of model training, internal testing, and external testing. Three different models were trained on 80% of the Linkou Chang Gung Memorial Hospital (LCGMH) dataset, and were tested on 20% of the LCGMH dataset for internal testing. The Kaohsiung Veterans General Hospital dataset was used for external testing. * Rule-based model was built at first round and tested in 10 rounds.

**Figure 3 diagnostics-12-01882-f003:**
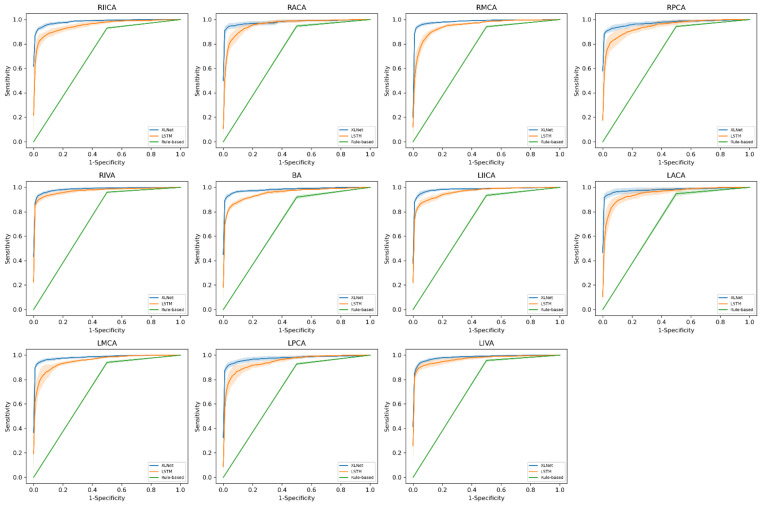
Comparison of the receiver operating characteristic curves (ROC) in stenosis detection tasks obtained by the rule-based model, the long short-term memory (LSTM) model, and the XLNet model with the internal test dataset. Each plot represents stenosis detection performance on each artery. The XLNet model clearly demonstrates superior performance with consistent larger area under the curve results compared with those of the LSTM model.

**Table 1 diagnostics-12-01882-t001:** The percentage of cases with confirmed stenosis (≥50% diameter stenosis) for each artery in both internal and external datasets.

	Internal Dataset (n = 9614)	External Dataset (n = 315)
RIICA (%)	740 (7.7)	12 (3.8)
RACA (%)	416 (4.3)	2 (0.6)
RMCA (%)	967 (10.1)	13 (4.1)
RPCA (%)	491 (5.1)	6 (1.9)
RIVA (%)	1052 (10.9)	2 (0.6)
BA (%)	554 (5.8)	9 (2.9)
LIICA (%)	735 (7.6)	4 (1.3)
LACA (%)	407 (4.2)	4 (1.3)
LMCA (%)	1005 (10.5)	10 (3.2)
LPCA (%)	547 (5.7)	3 (1.0)
LIVA (%)	943 (9.8)	2 (0.6)

BA—basilar artery; LACA—left anterior cerebral artery; LIICA—left internal carotid artery; LIVA—left intracranial vertebral artery; LMCA—left middle cerebral artery; LPCA—left posterior cerebral artery; RACA—right anterior cerebral artery; RIICA—right internal carotid artery; RIVA—right intracranial vertebral artery; RMCA—right middle cerebral artery; RPCA—right posterior cerebral artery.

**Table 2 diagnostics-12-01882-t002:** Comparison of area under the receiver operating characteristic curve (AUROC) results between internal and external testing datasets for the stenosis-identification task from three different NLP models. Results are expressed as mean ± standard deviation.

	Internal Testing Dataset (n = 1922)			External Testing Dataset (n = 315)		
Cerebral Artery (Prevalence in Internal/External Dataset) %	*Rule-Based Model*	*LSTM*	*XLNet*	*Rule-Based Model*	*LSTM*	*XLNet*
RIICA (7.7/3.8)	0.93 ± 0.01	0.95 ± 0.01	0.98 ± 0.00	0.71	0.76 ± 0.10	0.91 ± 0.11
RACA (4.3/0.6)	0.95 ± 0.01	0.96 ± 0.01	0.98 ± 0.01	0.50	0.73 ± 0.16	0.93 ± 0.01
RMCA (10.1/4.1)	0.94 ± 0.01	0.95 ± 0.01	0.99 ± 0.00	0.58	0.77 ± 0.10	0.97 ± 0.02
RPCA (5.1/1.9)	0.94 ± 0.01	0.95 ± 0.02	0.97 ± 0.01	0.50	0.58 ± 0.18	0.90 ± 0.06
RIVA (10.9/0.6)	0.96 ± 0.01	0.97 ± 0.01	0.99 ± 0.00	0.75	0.55 ± 0.19	0.99 ± 0.03
BA (5.8/2.9)	0.92 ± 0.02	0.95 ± 0.01	0.98 ± 0.01	0.83	0.47 ± 0.08	0.84 ± 0.04
LIICA (7.6/1.3)	0.93 ± 0.01	0.96 ± 0.01	0.98 ± 0.01	0.75	0.78 ± 0.07	0.93 ± 0.08
LACA (4.2/1.3)	0.95 ± 0.02	0.95 ± 0.01	0.98 ± 0.01	0.75	0.70 ± 0.15	0.99 ± 0.01
LMCA (10.5/3.2)	0.94 ± 0.01	0.95 ± 0.01	0.98 ± 0.00	0.50	0.80 ± 0.10	0.98 ± 0.01
LPCA (5.7/1.0)	0.93 ± 0.01	0.95 ± 0.02	0.98 ± 0.01	0.50	0.61 ± 0.14	0.79 ± 0.12
LIVA (9.8/0.6)	0.95 ± 0.01	0.97 ± 0.01	0.98 ± 0.00	0.50	0.65 ± 0.08	0.92 ± 0.09

BA—basilar artery; LACA—left anterior cerebral artery; LIICA—left internal carotid artery; LIVA—left intracranial vertebral artery; LMCA—left middle cerebral artery; LPCA—left posterior cerebral artery; RACA—right anterior cerebral artery; RIICA—right internal carotid artery; RIVA—right intracranial vertebral artery; RMCA—right middle cerebral artery; RPCA—right posterior cerebral artery.

## Data Availability

The data that support the findings of this study are available from Linkou Chang Gung Memorial Hospital and Kaohsiung Veterans General Hospital, but restrictions may apply to the availability of these data, which were approved by individual hospital IRB for the current study, and are not publicly available. However, processed datasets can be requested and made available from the authors with the permission from Linkou Chang Gung Memorial Hospital and Kaohsiung Veterans General Hospital.

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
