# Peer review of "Accurately Identifying Cerebroarterial Stenosis from Angiography Reports Using Natural Language Processing Approaches"

_diagnostics, 2022, doi:10.3390/diagnostics12081882_

Round 1

Reviewer 1 Report

This investigation aimed to evaluate the performance of 3 NLP models to assess for presence of intracranial artery stenosis in 11 vessels based on angiography reports. There was validation dataset using internal (20% remaining) and external data. The authors posit this technology can be used to risk-stratify patients, however admit challenges to model generalization in the context of non-standard reporting across institutions. Overall: please clarify purpose of paper beyond determining the best performing NLP model. Is it for when reports are needed to be incorporated to other risk stratification models (which ‘clinical applications’ as referred to in line 355)? Also, please state up front that comparing a rule/keyword based NLP system versus 2 deep learning based systems

Abstract: would clarify that the validation testing sets used both internal and external data

Introduction:

-Line 57: please include rate for Caucasian populations

-Line 58: ‘stroke recurrences were reported’ unclear intent of sentence? Are the rates for recurrence higher in asymptomatic individuals with IAS? What is timeline for this recurrence (ie would the NLP help for short term or long term risk stratification)

-Paragraph 2: clarity and editing necessary, many sentences are long and difficult to understand

Methods:

-the type of angiography for the LCGMH population needs to be specified. Is this CTA, MRA, other? This is especially relevant as reader wonders if training was on one modality, and external validation on another?

-how is the angiography preprocessing (line 99) accomplished? Has algorithm/code been previously described/published?

-The two chief neurologists reviewed only the 315 external dataset or all, meaning over 9,000 reports/exams? Unclear how arrived at the gold standard consensus and for how many of exams

Results:

-rule based performed better than LSTM for some vessels; please re-write/clarify results in lines 253 on

-Line 256 on is more suited to the discussion section

Discussion:

-why is there variable performance by vessel? Was there more inter-reader variability as well in vessels (if have access to this data from initial scoring). Were these vessels with poorer NLP performance different in any way (clinically/neurologically) in comparison to vessels in which NLP had better performance? Clinical opinion needed from the MD

-why was XLNet more robust in that maintained better performance on external data? Why may this model be more generalizable?

-why is there worse performance in left versus right vessels? MD input?

Tables:

--Table 1: what is the definition of ‘positive cases’? what degree of stenosis does this refer to?

Reviewer 2 Report

Dear authors,

Thank you for submitting this manuscript that is focused on the identification of eleven intracranial artery stenosis from angiography reports based on 3 different NLP classification algorithms.

The manuscritp shows the expected academic standards, it has clarity of presentation, well organised and clearly written with some minor typos.

Although the problem statetment is significant to the  body of knowledge, there is lacking some clinical relevance and in the Introduction section improvement can be made, namelly:

1. What is the real clinical problem? Does this mean that Radiologist; while developing the imaging reports from the angiogrtaphy exams do not develop a correct diagnosis, or that it is really difficult for other medical specialistis to understand it? Doesn't that imply a clinical malpractice?

2. How does the statement in line 67 to 69 - "It has been estimated that about (...) clinical applications" - is aligned with the problem of this work? The goal is to classify the results from the report of the angiography exams but not relate with the actual imaging analysis (which in this case is based on experts reading).

3. Please clarify what "clinical applications" are you referring to in the statement in line 71 to 73 and how it is related to the problem in study. 

The materials and methods section is well developed and the research methodology for the study is appropriate and applied properly. It would be interesting to see the results on a binary classification approach: stenosis versus non-stenosis approach and after evolve the classification of the artery related with the presence of stenosis - maybe for future studies. The strategy of using an internal DB and external DB is interesting and produce relevant results to the study.

The results are weel presented but its discussion is mainly focused on the rule-based model. The discussion section should be reviewed and more analysis to the XLNet model sould be included and in comparison to other research studyies. Also the clinical implications of the results should be discussed and the practicality of it.

The supproting evidence in this manuscript is relevant and strongly reliable.

Looking foward to see your work publish and future developments of this concept.
